# Validation of an orthotopic non-small cell lung cancer mouse model, with left or right tumor growths, to use in conformal radiotherapy studies

Li Ming Wang[1,2]*, Ranjan Yadav[3], Monica Serban[4,5], Osvaldo Arias[2],
Jan Seuntjens[4,5,6], Norma Ybarra[1,2,5]

1 Department of Experimental Medicine, Faculty of Medicine, McGill University, Montreal, Quebec, Canada,
2 Cancer Research Program, Research Institute of the McGill University Healthcare Centre, Montreal,
Quebec, Canada, 3 Medical Physics Unit, Department of Oncology, McGill University, Montreal, Quebec,
Canada, 4 Radiation Medicine, Princess Margaret Cancer Centre and Department of Radiation Oncology
and Medical Biophysics, University of Toronto, Toronto, Ontario, Canada, 5 Department of Oncology, McGill
University, Montreal, Quebec, Canada, 6 Department of Medical Biophysics, University of Toronto, Toronto,
Ontario, Canada

* li.m.wang@mail.mcgill.ca

pone.0284282

CHINA

**Data Availability Statement:** All relevant data are
within the paper and its Supporting information
files.

## Abstract

Orthotopic non-small cell lung cancer (NSCLC) mice models are important for establishing
translatability of *in vitro* results. However, most orthotopic lung models do not produce local-
ized tumors treatable by conformal radiotherapy (RT). Here we report on the performance of
an orthotopic mice model featuring conformal RT treatable tumors following either left or
right lung tumor cell implantation. Athymic Nude mice were surgically implanted with H1299
NSCLC cell line in either the left or right lung. Tumor development was tracked bi-weekly
using computed tomography (CT) imaging. When lesions reached an appropriate size for
treatment, animals were separated into non-treatment (control group) and RT treated
groups. Both RT treated left and right lung tumors which were given a single dose of 20 Gy
of 225 kV X-rays. Left lung tumors were treated with a two-field parallel opposed plan while
right lung tumors were treated with a more conformal four-field plan to assess tumor control.
Mice were monitored for 30 days after RT or after tumor reached treatment size for non-
treatment animals. Treatment images from the left and right lung tumor were also used to
assess the dose distribution for four distinct treatment plans: 1) Two sets of perpendicularly
staggered parallel opposed fields, 2) two fields positioned in the anterior-posterior and pos-
terior-anterior configuration, 3) an 180˚ arc field from 0˚ to 180˚ and 4) two parallel opposed
fields which cross through the contralateral lung. Tumor volumes and changes throughout
the follow-up period were tracked by three different types of quantitative tumor size approxi-
mation and tumor volumes derived from contours. Ultimately, our model generated deline-
able and conformal RT treatable tumor following both left and right lung implantation.
Similarly consistent tumor development was noted between left and right models. We were
also able to demonstrate that a single 20 Gy dose of 225 kV X-rays applied to either the
right or left lung tumor models had similar levels of tumor control resulting in similar adverse

**Funding:** Operating Grant Funding Program, which featuring funding from both the Cancer Research Society (https://www.societederecherchesurlecancer.ca/en) and the Cancer Institutes for Health Research (https://cihr-irsc.gc.ca/e/193.html), was received by Dr. Norma Ybarra (NY) under grant number 24224. The funders had no role in study design, data collection and analysis, decision to publish, or preparation of the manuscript.

**Competing interests:** The authors have declared that no competing interests exist.

outcomes and survival. And finally, three-dimensional tumor approximation featuring volume computed from the measured length across three perpendicular axes gave the best approximation of tumor volume, most closely resembled tumor volumes obtained with contours.

## Introduction

The use of pre-clinical orthotopic mouse models is essential for the development of novel therapies targeting solid tumors. Orthotopic models, despite being difficult and resource intensive to implement [1], allow for the development of a tumor situated in its origin micro-environment [2]. This results in a tumor model which features more representative tumor growth characteristics and response to realistic clinical treatment when compared to subcutaneous models featuring the same cancer cell lines [3, 4]. For research of non-small cell lung cancer (NSCLC), consideration of the tumor microenvironment is an important contributing factor to the progression and treatment response of solid tumors [5–9]. In radio-therapeutic research, the application of radiation to enhance immune response has meant that radiobiological research must take into consideration the surrounding tumor microenvironment when attempting to establish translatability of results [5]. In recent years, the orthotopic model has become a standard for establishing translatability or for validating *in vitro* results [10].

Despite its importance, the implementation of an orthotopic model is difficult and time consuming. Beyond just the technical difficulty of protocols featuring tumor implantation in the lung, which requires surgical implantation, tumor tracking over time is also challenging. The use of bioluminescence (BLI) imaging featuring the implantation of luciferase transfected cancer cell lines allows for the quantification of tumor size by correlating tumor size with signal intensity [11–13]. However, BLI imaging is highly affected by light scattering, tissue penetration, and absorption making quantification of absolute tumor sizes difficult, allowing only for relative quantification [11] and is not appropriate in applications where precise tumor delineation and visualization is required. Specifically, BLI is not appropriate in applications such as external beam radiotherapy, where treatment planning requires clear 3D reconstructions of tumor targets and nearby organs of interest. In addition, the expression of tdTomato fluorescent protein labeling and luciferase in a Lewis lung carcinoma has been shown to alter the tumor microenvironment through increasing the presence of tumor-infiltrating lymphocytes in the tumor [14]. The modification of the microenvironment could pose a potential confound for studies investigating radiation-induced microenvironmental changes or immunogenic outcomes of radiation. Most relevant for immunocompromised mice is the potential effect of microenvironment modifications on innate immune responses, as certain immunocompromised nude mice strains still have residual immune cell populations despite the loss of T-cells central to their adaptive immune systems [15]. As such, computed tomography (CT) imaging, is an alternative imaging modality, allowing for more accurate tumor delineation for treatment planning, validation of implantation success, and is suitable for precise longitudinal monitoring of tumor volumes [16].

Currently, there are a large variety of protocols for inducing an orthotopic model of NSCLC tumors in mice. Some of these proposed protocols feature left [12, 17, 18] or right [11, 13, 19, 20] lung transthoracic injection-based implantations which involves the delivery of cancer cells through the chest wall, with or without dissection of the superficial and/or chest wall

fascia. Two published protocols specifically require invasive dissection of the intercostal muscles to expose the lungs prior to injection [11, 19]. As an alternative to these surgical protocols there is also an intratracheal method which involves directly depositing the cancer cells into the lungs through the airways while the animal is intubated and anesthetized [3, 19, 21–23]. Despite the large variety of published models, there is a lack of information on the performance of the methodologies, whether they result in reproducible orthotopic tumors, and the ability for these orthotopic model to generate tumors with clear boundaries appropriate for imaging-based quantitative outcomes in radiotherapeutic or radiobiological research. Furthermore, despite the anatomical differences between the left and right mouse lung, left or right lung tumor models are used and reported on interchangeably. Currently, there is no validation performed of the possible different outcomes of a left or right lung tumor model. Despite the radical difference in anatomy possibly contributing to differences in implantation outcomes, tumor growth patterns and/or RT treatment response due to different organ avoidance constraints. It is also unclear if left and right lung models can generate tumors that could be treated with conformal radiotherapy (RT). Lastly, there is scarce literature specifically evaluating the characteristics of orthotopic lung tumor models featuring a p53 deficient H1299 NSCLC cell line implanted in either the left or right lung.

In this study, we report on the efficacy of an orthotopic mouse model of NSCLC using the p53 deficient H1299 NSCLC cell line. We assessed the outcomes of both a left and right lung orthotopic model and validate if there are differences in tumor implantation and development between left and right lung models. In addition to validating our protocol for both left and right tumor models, we assessed the dose characteristics for four distinct radiation treatment plans, in both the left and right lung, for the delivery of a single 20 Gy dose of 225 kV X-rays. We then also assessed *in vivo* tumor control of a parallel opposed treatment plan in left lung models and a more conformal four beam treatment plan in the right lung tumor models. Lastly, in efforts to optimize tumor quantification in follow-up imaging, we assessed and validated the use of different CT-imaging based tumor quantification methods to help establish best practices for longitudinal tumor quantification.

## Materials and methods

### Ethics statement

All experiments were approved by the Animal Care Committee at the Research Institute of the McGill University Health Centre (RI-MUHC) and in accordance with the ethical guidelines of the Canadian Council on Animal Care (Protocol Number: 8117). Athymic Nude Crl:NU (NCr)-Foxn1$^{nu}$ mice (Charles River Laboratories, Canada) used for experiments were housed at the Animal Resources Division of the RI-MUHC. Animals were housed maximum of 5 per cage and fed water and standard mice kibble *ad libitum*. All surgeries were performed with animals anesthetized with 2% isoflurane gas and with subcutaneous injection of slow-release buprenorphine as an analgesic. Animals were euthanized at 30 days post-treatment administration or if the animals reached the humane endpoint. Euthanization was conducted using 5% isoflurane anesthetization, followed by carbon dioxide asphyxiation and cervical dislocation. All efforts were made to minimize the suffering of the animals.

### H1299 culture and preparation

NCI-H1299 NSCLC cells were cultured in RPMI medium supplemented with 10% fetal bovine serum. Cells used for injection were not passaged more than three times to maintain similar growth characteristics among preparations [24].

On the day of orthotopic implantation, an hour prior to surgery, H1299 cells were trypsinized and washed in PBS. Cells were counted, and 400,000 H1299 cells were re-suspended in a mixture of ice-cold serum free RPMI, Matrigel (Corning Life Sciences, USA) and Omnipaque 350 (GE Healthcare, USA) in a 2:2:1 ratio, in a total volume of 20 uL. The combined volume was loaded into a 0.3 mL insulin syringe (BD, USA) and kept on ice until use.

## Orthotopic H1299 cell implantation

Animals were given slow-release buprenorphine during the morning of surgery at a dose of 1.0 mg/kg. At the start of surgery, animals were induced with 2.5% isoflurane gas and maintained at 2% throughout the procedure. Animal reflexes and adequate anesthetic depth were verified through pinching of palm and foot pads. For injections in the left lung, mice were placed in the right lateral decubitus position to expose the left middle axillary line. For injections in the right lung, mice were placed in the left lateral decubitus position exposing the right middle axillary line. Once the middle axillary line was located, on either the left or right side, a 1–1.5 cm incision in the cranial-caudal direction was made at the middle axillary line at the level of the chest wall centering on the lower third of the lung. Tissue and fascia were dissected using forceps and kept open with a wire speculum to visualize the chest wall and the movement of the lung. The syringe containing the cell preparation was perpendicularly inserted between the third intercostal space, counting from the lowest true rib towards the cranial direction (Fig 1), at the region that is intersected by the intercostal space and the middle axillary line to a depth of 4 mm (Fig 2). If the bottom border of the lung was visualized to be one intercostal space away from injection site during inhalation, the injection site was moved towards the next intercostal space in the cranial direction. Contents of the syringe were pushed into lung slowly to avoid reflux of material into thoracic cavity. After complete deposition of material, syringe was kept in place for another 20 seconds to allow solidification of Matrigel. The syringe was then extracted, fascia closed, and skin sutured with vinyl sutures. Animals were allowed to recover from anesthesia on a heating pad, and the success of the surgery was verified by CT imaging.

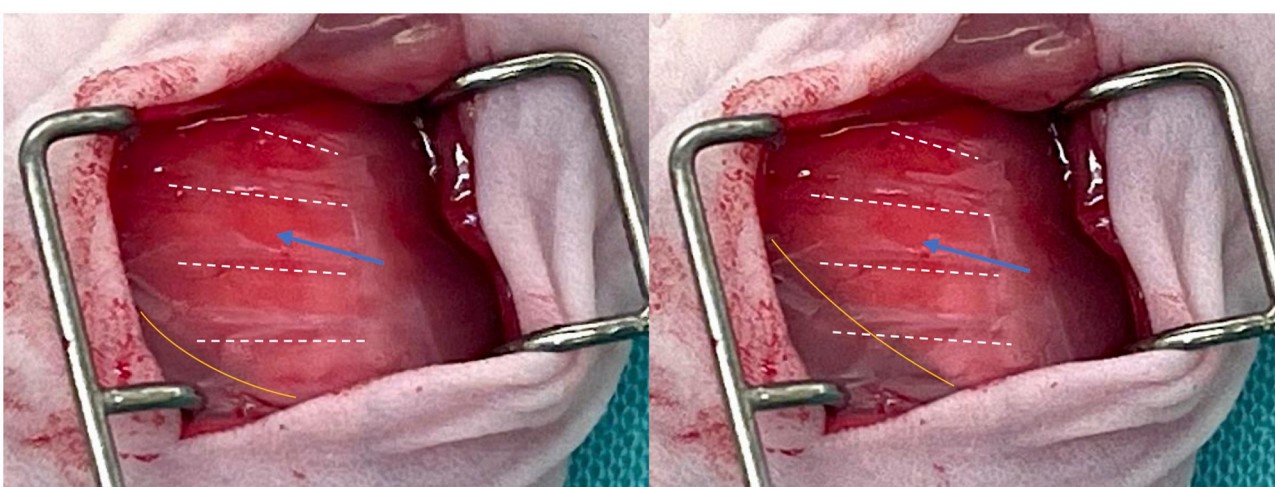

**Fig 1. Image of the incision site featuring the lung visible through the chest wall.** Images show a lung on exhale (left) and inhale (right). Relevant anatomical landmarks such as the ribs (white dashed lines), lower border of the lung (orange line) and the third intercostal space (blue arrow) counting from the lowest true rib towards the cranial direction are highlighted.

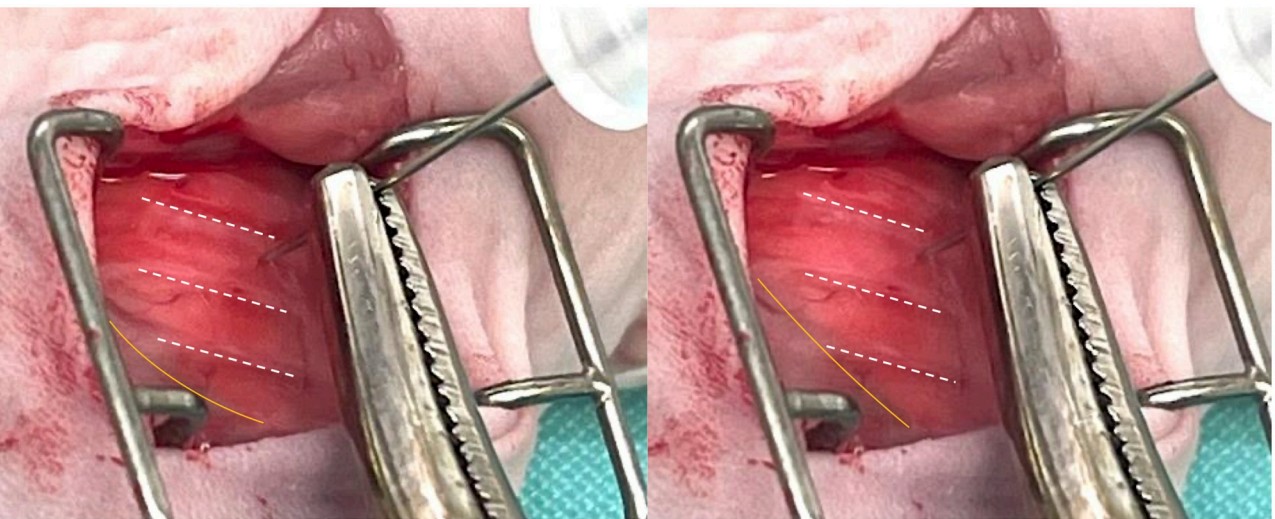

**Fig 2. Image of the incision site featuring the lung visible through the chest wall.** Images show a lung on exhale (left) and inhale (right). Syringe is advanced to a depth of 4 mm, measured from tip of bevel to lung surface. While not shown in this image for clarity, locking forceps were fixed 4 mm from the tip of the needle to impede the syringe from being inserted further than the acceptable depth.

## Cone beam CT imaging

Immediately after surgery, an initial cone beam CT (CBCT) image was taken to assess the deposition of the injected cells. Animals were induced to anesthesia using 2.5% isoflurane and maintained on 2.5% isoflurane throughout the duration of imaging. Animals were placed in prone position, head-first on the carbon fibre couch of the small animal irradiator (SmART+) XRAD system (Precision X-Ray Inc., Madison, CT, USA). A 3 mm aluminium filter was used in combination with a low dose imaging protocol which featured a 40 kV and 3 mA x-ray beam and reconstruction of the volume was with an isotropic 0.1 mm by 0.1 mm by 0.1 mm voxel resolution. Imaging was acquired within 4 minutes including the entirety of the thorax and abdomen from the base of the skull to the pelvis. Success of the injection was verified by the appearance of contrast medium within the lung parenchyma during CT imaging (S1 Fig—images in the first column from the left). Imaging was saved using calibrated parameters featuring a slope value of 2584 and an intercept value of -1000 to ensure that CBCT imaging contains correct density values. After imaging, animals were allowed to recover from anaesthesia in a separate chamber and then returned to normal housing.

## Tumor growth and animal follow-ups

Animals were followed-up by imaging every three and a half days, bi-weekly, using the same imaging procedure and parameters as mentioned above. Images were then exported from the SmART+ software (Precision X-Ray Inc., Madison, CT, USA) and opened using 3D Slicer [25]. During these follow-ups, animal weights were recorded, and an assessment of the animal's overall well-being was conducted. This included identifying signs of distress and symptoms associated with NSCLC lung tumors, such as dyspnea, rapid weight loss, and traditional signs of murine distress (such as weakness, lethargy, reduced grip, reduced activity, and lack of grooming). Animals were followed-up for 30 days after radiation treatment. However, animals were euthanized prior to the 30 days if they demonstrated moderate to severe signs of distress, weight loss, and/or tumors larger than ½ of the volume of the lung.

## Dose calculation and radiation treatment

Treatment planning and dose calculations were conducted on a representative tumor bearing animal to validate a generalizable treatment that can be applied to all animals. Animals that were identified with tumors of appropriate size, between 5–20 mm$^3$ were selected and imaged according to imaging protocol outlined for animal follow-up imaging. Animal imaging was then exported from the SmART+ software in standard DICOM format to the SmART-ATP Treatment planning software (Precision X-Ray Inc., Madison, CT, USA). The planning software uses the dose calculation engine DOSXYZnrc based on the EGSnrc Monte Carlo system. Since a 3D Monte Carlo dose calculation in the kV energy range depends critically on the tissue composition, assignment of tissue composition was established into four different material categories. International Commission on Radiation Units and Measurements (ICRU) report 44 [26] standard medium compositions were used in the dose calculation algorithm. The different media representing tissue classes included: air—AIR521ICRU, lung—LUNG521ICRU, soft tissue TISSUE521ICRU, and bone—BONE521ICRU. These different media were delineated on the imported CT to allow the planning software to identify the appropriate material composition of the structures in the CT image. Organs at risk (OAR) and the tumor (target) were then contoured also contoured on the same planning software. Fields of 5 mm diameter were placed in four canonical configurations for both left and right tumor models (Fig 3): (1) Two sets of perpendicular staggered parallel opposed fields with avoidance of both spinal cord and heart (Fig 3 –Row 1 & 2), (2) two fields positioned in the anterior-posterior and posterior-anterior (APPA) configuration (Fig 3 –Row 3 & 4), (3) an 180˚ arc field from 0˚ to 180˚ (Fig 3 –Row 5 & 6) and (4) two parallel opposed fields which crosses through the contralateral lung as well as the target with avoidance of both heart and spinal cord (Fig 3 –Row 7 & 8). All plans were calculated to deliver 20 Gy in one fraction to 100% of the volume of the target. After the field parameters were established, plans were calculated using particle density of 550,000 histories per mm$^2$ field area with an isotropic voxel resolution of 0.1 mm x 0.1 mm x 0.1 mm. Once the calculation was completed, contours, field plans and dose statistics were exported and saved. The dose of 20 Gy in a single fraction was selected as similar doses were reported to be well tolerated in small rodents [27–29].

Two of the four treatment plans were delivered in effort to assess *in vivo* tumor control. For right lung lesions, the treatment plan featured the 4-field with 2 sets of parallel opposed fields oriented in a cross-pattern (Fig 3 row 1). Alternatively, for the left lung lesions, the treatment plan featured was the 2 parallel opposed fields which crosses through the contralateral lung (Fig 3 row 8). These two plans were chosen as the respective right and left lung plans as they provided the most conformal field orientation which offered the best avoidance of the heart and spinal cord (Fig 3 row 1 and Fig 3 row 8). With the heart being present more in the left side of the thorax, the 4-field plan enabled additional conformality. For RT delivery, animals were first imaged through the same imaging protocol outlined for animal follow-up imaging. After imaging was acquired, the mice were maintained on 2.5% isoflurane while the imaging was exported from SmART+ and uploaded to the SmART-ATP treatment planning software. In the interest of time, only the delineation of structures and the placement of fields were conducted. Four different material types were set following the same method as previously described under the section outlining treatment plan calculation. Fields were placed in the 4-field and 2 parallel-opposed field configurations for right and left tumor models respectively while best avoiding OARs. Plan calculations were conducted using similar particle density and isotropic voxel spacing as previously mentioned. Treatment protocol is exported from the SmART treatment planning software and delivered by the SmART+ software using a 225 kVp and 13 mA x-ray field with a 0.3 mm Cu filter along with a 5 mm cylindrical collimator.

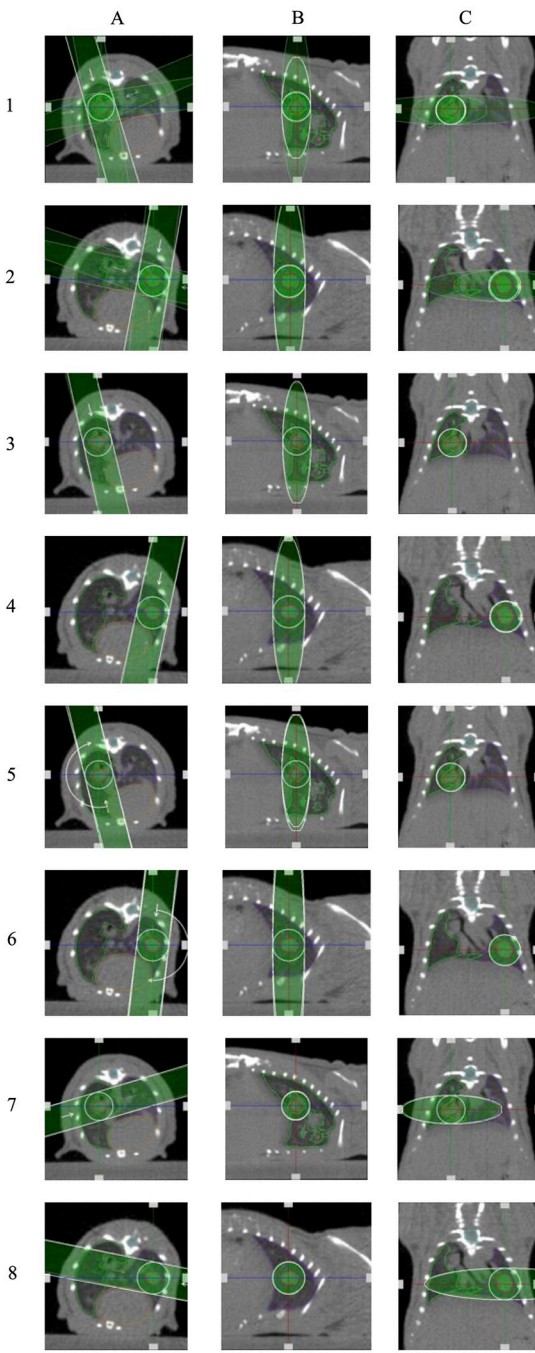

**Fig 3. Treatment plans for both the left and right lung.** The columns are views of the treatment plan from three visual axes, transversal (Column A), sagittal (Column B) and coronal (Column C). All the even numbered rows are plans which featured left lung lesions while all odd numbered rows are plans featured right lung lesions. The plans shown are 1) 2 sets of perpendicular staggered parallel opposed fields with avoidance of both spinal cord and heart (Row 1 & 2), 2) 2 fields positioned in APPA configuration (Row 3 & 4), 3) an 180˚ arc field from 0˚ to 180˚ (Row 5 & 6) and 4) 2 parallel opposed fields which crosses through the contralateral lung as well as the target with avoidance of both heart and spinal cord (Row 7 & 8).

## Tumor quantification

Tumors were quantified through three main metrics of measurement based on the bi-weekly follow-up CT imaging: 1) unidimensional measure of the longest diameter based on the Revised Response Evaluation Criteria in Solid Tumors (RECIST) [30], 2) a two-dimensional conventional tumor volume calculation featuring only the longest axes and the smaller of the two perpendicular axes [31], and 3) a three-dimensional estimation of ellipsoidal volume based on three perpendicular axes [31]. Relevant tumor measurements were made in the 3D Slicer [25] system after images were uploaded and reviewed. Unidimensional measure consisted of the longest diameter of the tumor visualized in any of the three planes of the reconstructed CT imaging (Fig 4). The two-dimensional tumor volume calculation, which was traditionally used for

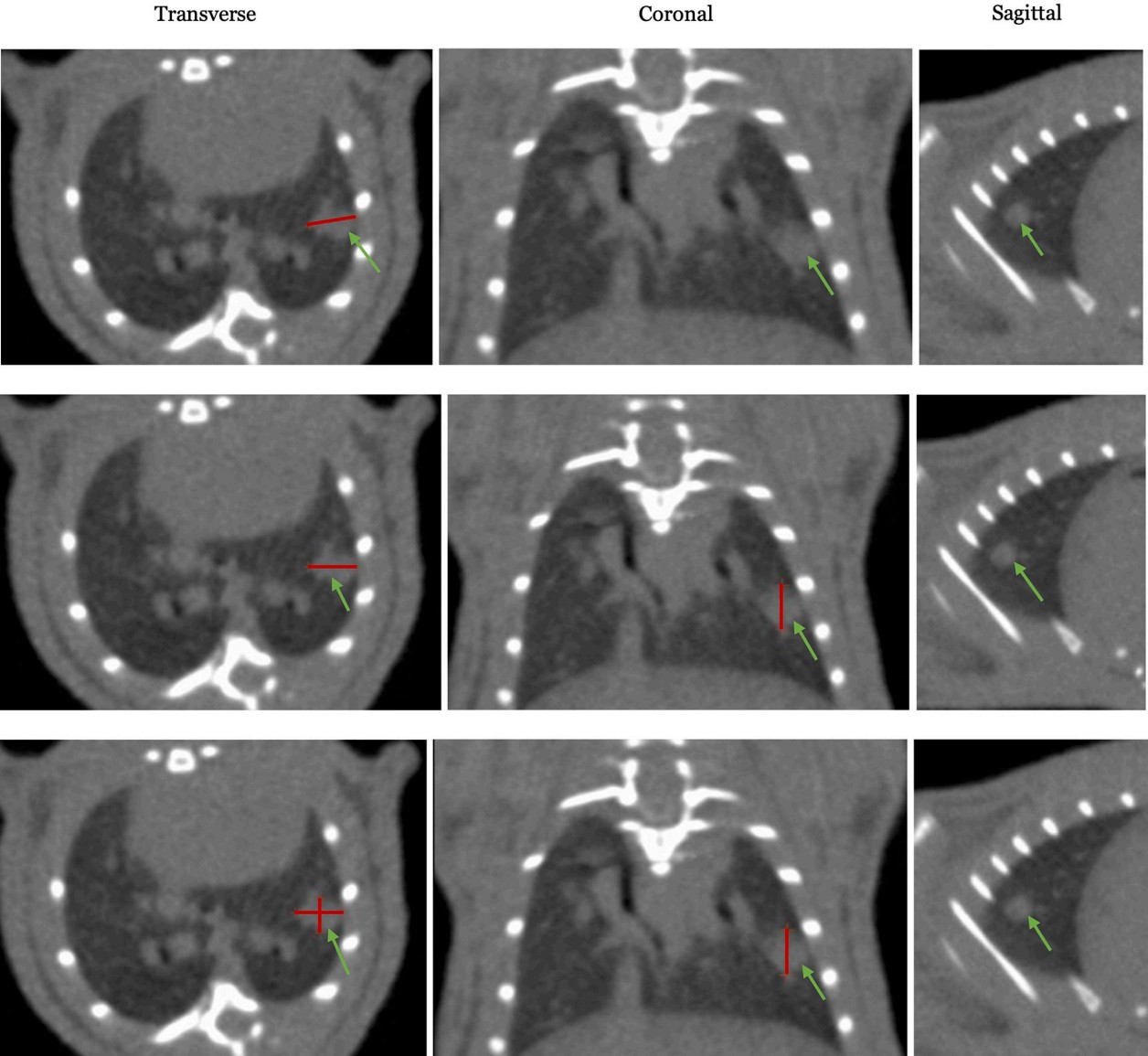

**Fig 4. The images feature the transverse (1st image of every row), coronal (2nd image of every row), and sagittal (3rd image of every row) slices of the CT image at the location of the lesion (indicated by the green arrow).** Each row of images features methods of measuring (as delineated by red lines) based on the RECIST criteria (1st row from the top), two-dimensional traditional measurement (2nd row), and three-dimensional ellipsoid-based measurement (3rd row).

caliper-based tumor measurements of subcutaneous tumors, was calculated as:

$$V_{2D} = \frac{W_{2D}{}^2 * L_{2D}}{2}$$

wherein $W_{2D}$ was identified as the tumor width and $L_{2D}$ as the tumor length [31]. In our study, $W_{2D}$ was identified as the widest diameter in the transverse view, while $L_{2D}$ was the length of the tumor in the coronal view. The three-dimensional analysis was calculated using the equation:

$$V_{3D} = \frac{4}{3}\pi \left(\frac{W_{3D}}{2}\right)\left(\frac{L_{3D}}{2}\right)\left(\frac{H_{3D}}{2}\right)$$

wherein $W_{3D}$, $L_{3D}$, and $H_{3D}$ were the diameters of the lesion along the normal X, Y and Z-axis, respectively, of the tumor as obtained from the transversal and sagittal views.

These three methods of measurement were compared to a contouring-based volume measurement. The contour-based measurement was generated by exporting and uploading CT imaging into the Varian Eclipse system (Varian Medical Systems, USA) where the gross tumor volume was manually contoured as a region of interest (ROI). The gross tumor volume structure was saved and exported as a RTstruct DICOM file and imported into 3D Slicer [25]. Within 3D Slicer, a binary mask of the ROI was constructed using the RTstruct file and the subsequent conversion was saved as an NRRD file. Using the NRRD file, the volume of the ROI, denoted by $V_{voxel}$, was approximated by multiplying the number of voxels ($N_v$) in the ROI's binary mask by the volume of a single voxel.

## Animal euthanasia and visual tumor identification

After 30 days of follow-up or when animals have reached their humane endpoint, animals were sacrificed. During the euthanasia process, animals were heavily sedated using 5% isoflurane, the chest wall was opened with diaphragm resected to expose the thoracic organs. Perfusion of the lungs was conducted by advancing a 25-gauge syringe loaded with 10 mL of physiological saline through the apex of the heart into the left ventricle. Prior to injection of the saline, an incision was made on the right atrium to allow flow of blood. Saline was pushed until flow from atrial incision became clear, both lungs became white, and the animal has expired. Once perfusion is completed, the heart is removed via excision at the root and thoracic cavity is examined for instances of tumor growth within the lung parenchymal and instances of lung external lesions. Instances were logged and recorded.

## Statistical analysis

Statistical analysis was conducted using GraphPad Prism 9 for Windows (GraphPad Software, LLC). Normality for all data sets was computed using the Normality and Lognormality test packages. Significant differences between the tumor volumes of CTRL and RT treatment groups were calculated using two-tailed nonparametric Mann-Whitney test to derive p-values. Spearman's correlation was calculated to compare the correlation between the three different modalities of tumor volume measurements to evaluate which of the three different dimensional estimates of tumor volumes performed the best by being the most in accordance with contoured volumes.

## Results

### Orthotopic tumor development

The implanted H1299 cell line produced steadily growing singular nodules after 2 to 3 weeks of growth in both left and right lung implantation models. Prior to the visible growth of

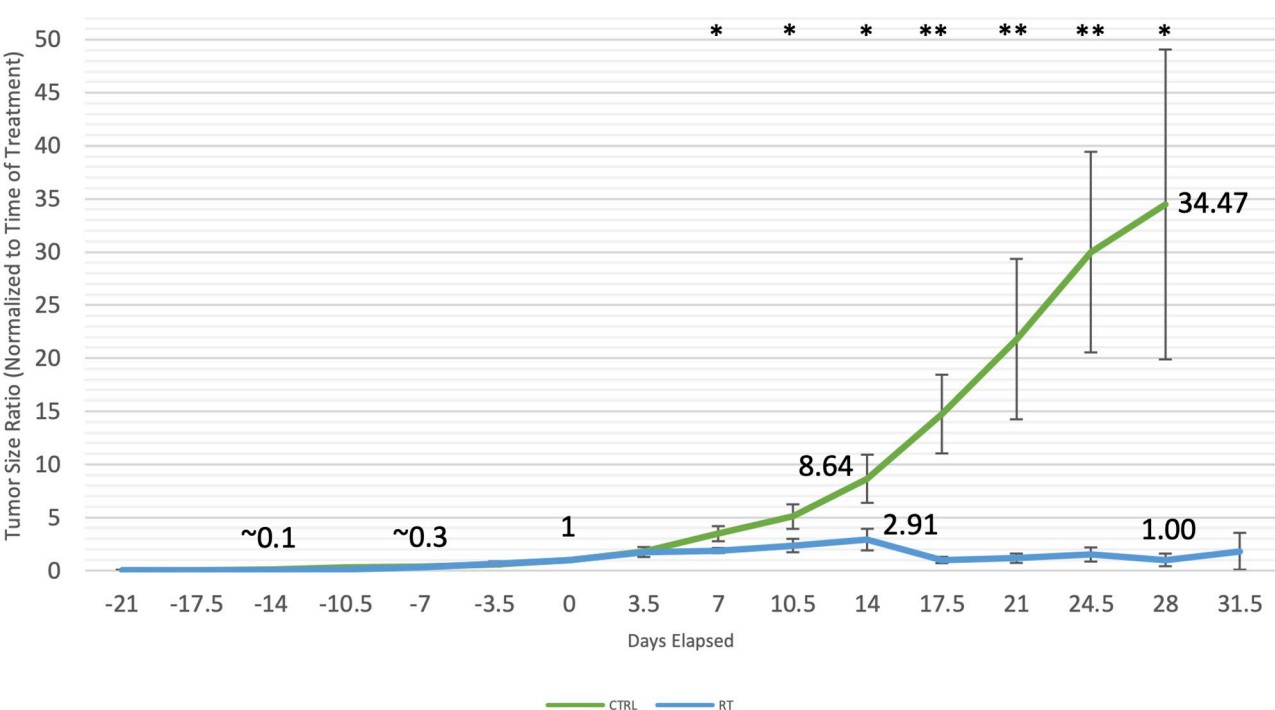

**Fig 5. Graph of tumor response in CTRL and RT animals.** Graph contains both left and right implanted tumor responses as they performed similarly. Tumor volumes normalized to tumor volume at the time of treatment are aligned based on date of sham RT or RT treatment (Day 0). Animals were separated into non-treated (CTRL—green line) and RT treated (RT—blue) with 14 and 13 animals in each group, respectively. The asterisk indicates the p value of the computed Mann-Whitney test as being $p < 0.05$ (*) or $p < 0.005$ (**) between the CTRL and RT groups.

tumors, approximately half of all animals exhibited CT densities local to the site of injection, potentially due to acute inflammation or residual contrast medium. This early period takes anywhere from 1 to 2 weeks to fully resolve and can be distinguished retroactively from the period of true tumor growth. Characteristics of this early period are an area of high Hounsfield density, similar to soft tissue density, shrinking over a period of 1 to 2 weeks and often sharper borders than that of the tumor which eventually develops (S2 and S3 Figs). This initial period of non-tumor soft tissue density eventually transitioned into confirmable tumor density, albeit with overlap between disappearance of the density and appearance of the retroactively confirmable tumor density. These confirmable tumor densities are characterized by consistent growth and can be observed in both right and left lung tumors. When subjected to RT treatment, significant tumor control was observed (Fig 5). Overall, we achieved an implantation success rate of 80% for both right and left lung implantations, of which all successfully implanted animals resulted in RT treatable tumors following our protocol. All tumors eventually reached treatable volumes within 2 to 3 weeks following implantations. CT image findings of tumors were consistent throughout the time of follow-up with bi-weekly imaging capable of tracking tumor growth.

Despite similar tumor growth characteristics and treatment outcomes between animals featuring right and left tumors, 10 in 27 animals that had right lung injections were found to have external lung lesions in the thoracic cavity (Fig 6) at the endpoint. However, in comparison, left lung injected animals only resulted in external lung lesions for every 5 in 27 animals. It is

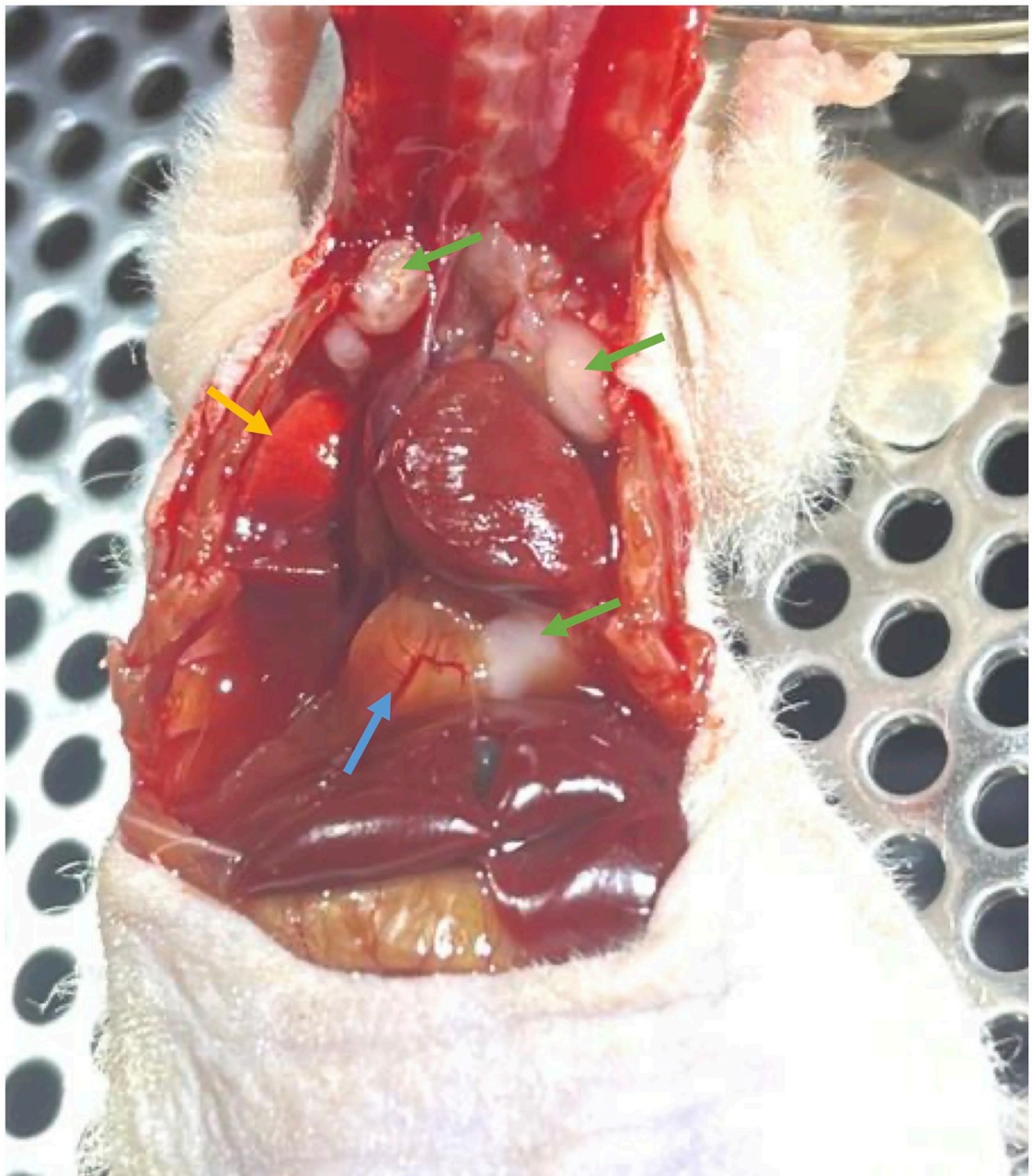

**Fig 6. Examples of lung external tumors located in the thoracic cavity.** Tumors are indicated with the green arrows while the lung is indicated with the yellow arrow. The blue arrow indicates the diaphragm, on which the lowest of the three indicated lesion sits. This example animal is shown pre-perfusion during necropsy with multiple lung external lesions visible in the thoracic cavity. The mouse was injected with tumor cells into the right lung as per the methodology outlined.

unclear whether these lesions were because of the initial injection being partially refluxed from the lung and being deposited into the thoracic space or was a result of metastatic spread. The multi-lobule structure of the right lung could explain the increased prevalence of external lung tumor in the right. The lobes of the right lung are small, leading to increased likelihood of reflux due to reduced volume. Furthermore, the syringe bevel, being approximately 1 mm in

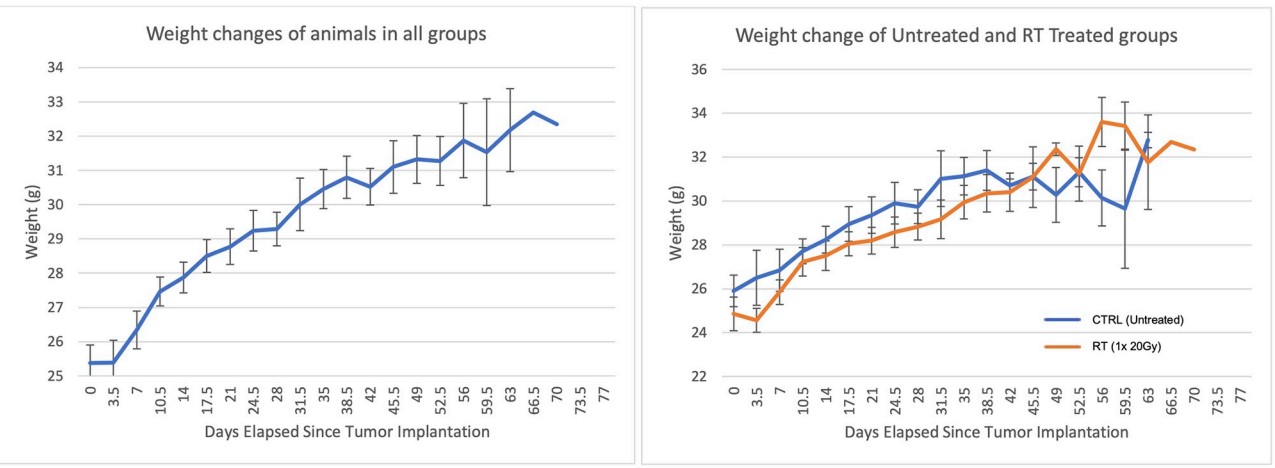

**Fig 7. Average animal weights across all groups post-surgery (left), n = 27, and animal weights as divided by treatment group (right) showing CTRL group animals (blue), n = 14, and RT animals given one dose of 20 Gy (orange), n = 13.** Both left and right lung implanted animals responded similarly and were grouped together for analysis. Days elapsed refers to the number of days since initial tumor cell implantation.

length, can be positioned only partially in a one of the three right lobes despite having reached appropriate depth. Depositing part of the implanted cells into the fissures between the lobes, which led to greater incidence of tumors external to the lung. These external tumors did not affect the tumor growth rate and tumor response to RT that we observed.

All animals tolerated the treatment well, with no acute side effects. Throughout the experimental period, animals had consistent weight gain. Animals had an average of 25.4±0.5 g when first received and reached an average maximum of 32.2±1.2 g (Fig 7). There was an initial one-week stagnation in growth or decrease in weight due to post-surgical recovery. This effect was observed in both CTRL and RT groups.

After 4 weeks of post-surgical growth, weights were particularly indicative of animal condition, with >1 g of weight loss every 3 days coinciding with persistent distress related behaviours and tumors approximating two-thirds to half the volume of the lung. This rate of loss was considered rapid weight loss and was related to the mice approaching the humane intervention points for rodent cancer models.

Animals survived on average 18 days after treatment, for both left or right tumor implantation (Fig 8), regardless of treatment group.

## Dose deposition of different treatment plans

Animals which developed tumors satisfying the volume criteria were used to generate treatment plans. Contouring of the left lung, right lung, and spinal cord were trivial. However, due to the animal's anatomy, the small size of the animal, and the resolution of the CT, the border between the thymus and heart was visually indistinct. Hence delineation of the superior border of the heart was approximate (S4 Fig). Esophagus was also not delineated despite its clinical importance due to an inability to differentiate the esophagus from nearby soft tissue structures and image resolution. The dose-volume histograms (DVHs) for the four types of treatment plans and for both left and right lung lesions are presented in Fig 9 along with relevant dose values in Table 1. Tumors in the left lung resulted in greater deposition of dose in the surrounding OARs due to both the smaller volume of the lung and the heart taking up space in this part of the thorax. While AP-PA fields can traverse the right lung without crossing OARs,

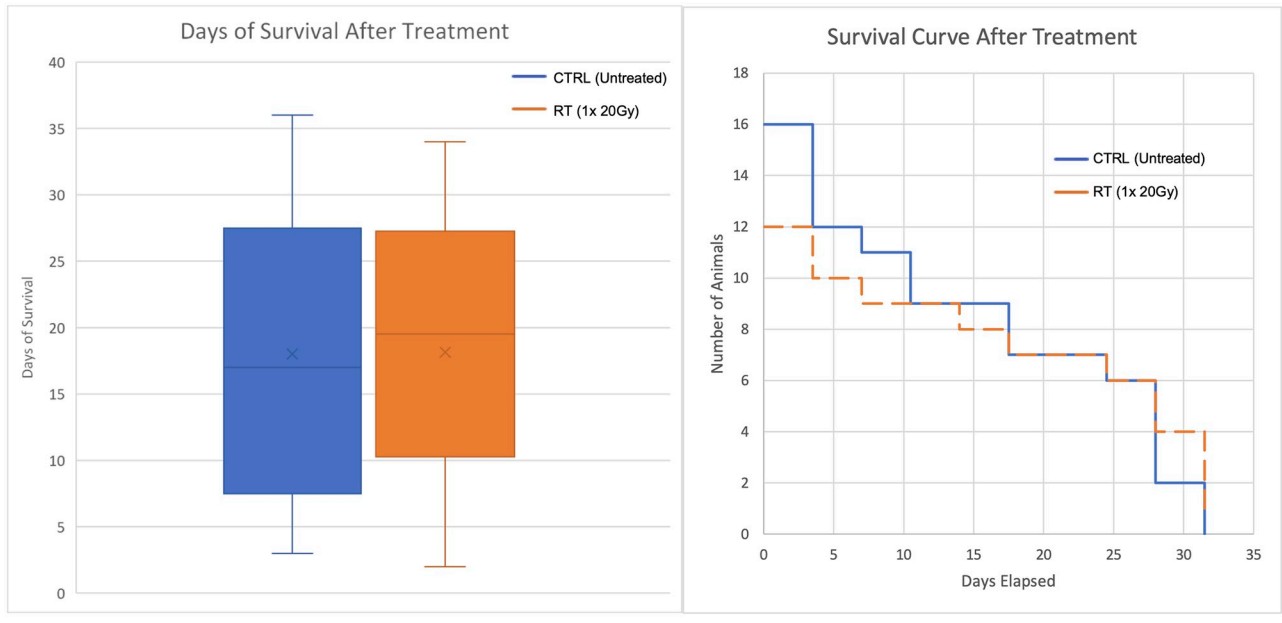

**Fig 8. Plots featuring survival of CTRL and RT animals.** Box and whisker plot showing animal survival after RT (left), n = 27, comparing survival of untreated animals (blue), n = 14, and animals given a single dose of 20 Gy (orange), n = 13. Kaplan-Meier survival curve showing the rate of animal mortality throughout the period after RT (right), n = 28, featuring survival of untreated animals (blue), n = 16, and animals given a single dose of 20 Gy (orange), n = 12. Like tumor size response, both left and right lung animals had similar survival outcome and were grouped together for analysis. The same treatment regimen is applied to animals in Fig 5 and the current figure.

in the left lung, AP-PA fields must traverse portions of the left ventricle and left atrium. Similar interaction with the heart occurred for the arc and parallel perpendicular plans. The parallel opposed field plan which features crossing the contralateral lung has the potential to avoid both the heart and spinal cord but does so through deposition of the entire treatment dose to significant portions of both the contralateral lung and the mediastinum. Despite these reported doses to the heart and spinal cord in the cases animals had not exhibited any acute toxicities or reactions to RT.

## Comparing methods for tumor tracking

When compared to contoured volumes, to assess the ability of dimensional approximation to capture tumor volumes, the 3D analysis had a Spearman' r score of 0.9902 (p<0.0001), the 2D analysis an r score of 0.9897 (p<0.0001), while the RECIST single dimension measurement had an r score of 0.9876 (p<0.0001). This indicates that all three methods of tumor volume quantification can capture the changes in tumor volume almost equally well. However, when the absolute volume approximation of the 3D and 2D calculation are compared to the contoured volumes from a follow-up-to-follow-up basis, there is consistent over-estimation as measured dimensions are reduced. With 2D measurements being the most likely, 12 out of 13 samples, to overestimate volumes. While 3D measurements were able to more closely approximate contoured volumes than 2D measurements (Fig 10).

## Discussion

In this study we reported on the performance of an orthotopic mouse model of NSCLC featuring non-invasive CT imaging for tumor quantification for both right and left lung models. This model was effective at generating tumors appropriate for conformal beam RT and for

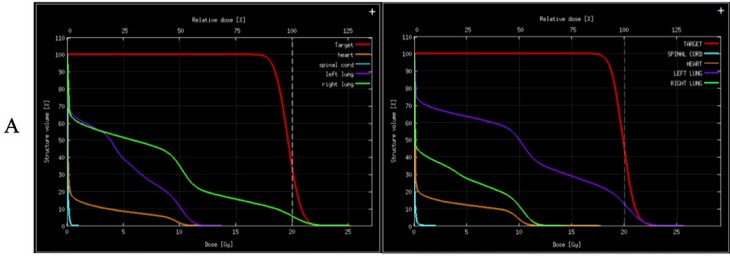

Two sets of perpendicular staggered parallel opposed fields with avoidance of both spinal cord

and heart. Plan depicted in Figure 3 Row 1 (right lung) and Figure 3 Row 2 (left lung).

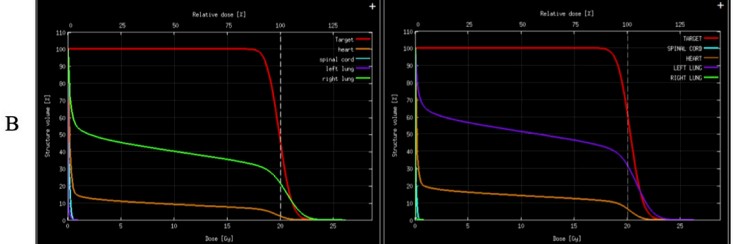

Two fields positioned in the AP-PA configuration. Plan depicted in Figure 3 Row 3 (right lung)

and Figure 3 Row 4 (left lung).

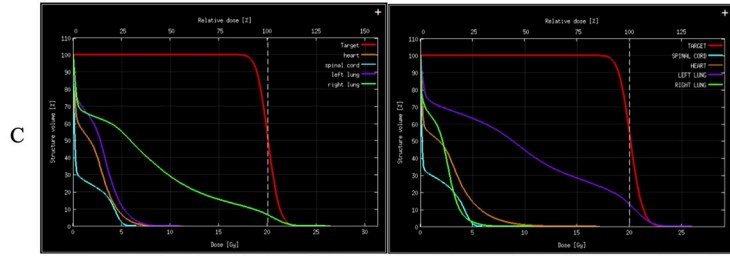

180º arc field from 0º to 180º. Plan depicted in Figure 3 Row 5 (right lung) and Figure 3 Row 6

(left lung).

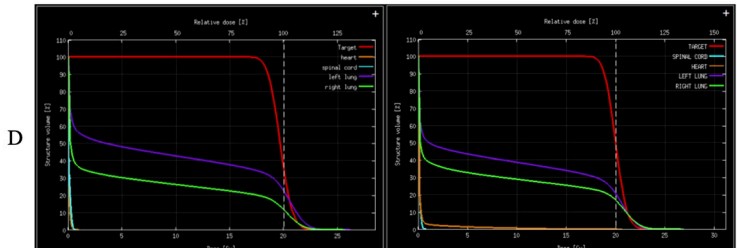

Two parallel opposed fields which crosses through the contralateral lung as well as the target

with avoidance of both heart and spinal cord. Plan depicted in Figure 3 Row 7 (right lung) and

Figure 3 Row 8 (left lung).

**Fig 9. DVHs of different treatment plans.** Consistent across all DVHs are dose characteristics of the target, delineating the gross tumor volume (red), the heart (orange), spinal cord (cyan), purple (left lung) and green (right lung). The DVHs were separated into right lung (left column) and left lung (right column) DVHs. With each row representing the different treatment plans.

precise quantitative tumor tracking of dynamic tumor response. Our model was able to demonstrate that the H1299 cell line generates consistent singular nodules, when surgically implanted, validating previously published protocol [18], and can be injected into either the left or right lung to yield singular nodules appropriate for conformal external beam RT

**Table 1. Dose values, in Gy, for all four treatment plans including: 1) two sets of perpendicular staggered parallel opposed fields with avoidance of both spinal cord and heart (denoted 4 Field), 2) two fields positioned in the AP-PA configuration (denoted AP-PA), 3) two parallel opposed fields which crosses through the contra-lateral lung as well as the target with avoidance of both heart and spinal cord (denoted Lat. Opp.), and 3) an 180˚ arc field from 0˚ to 180˚ (denoted Arc).** The mean dose (Dmean), dose to 95%, 80%, 50%, 20%, 5% and 1% of the structure volume are listed (rows) for each of the five contoured structures. Volumes for both the representative right and lung implanted lesion, for which the plans were generated, and values were calculated for, were 8 and 7 mm$^3$ respectively.

| | | Right Lung Lesion | | | | Left Lung Lesion | | | |
|---|---|---|---|---|---|---|---|---|---|
| | | 4 Beams | APPA | Lat. Opp. | Arc | 4 Beams | APPA | Lat. Opp. | Arc |
| GTV | Dmean | 19.9 | 19.9 | 20.2 | 20 | 19.9 | 19.9 | 20 | 20.1 |
| | D95% | 19.2 | 18.9 | 19.1 | 18.6 | 19.2 | 18.9 | 19 | 18.8 |
| | D80% | 19.5 | 19.4 | 19.6 | 19.3 | 19.6 | 19.4 | 19.5 | 19.4 |
| | D50% | 19.9 | 19.9 | 20.1 | 19.9 | 19.9 | 19.9 | 20 | 20.1 |
| | D5% | 20.8 | 21.1 | 21.3 | 21.4 | 20.6 | 20.9 | 21 | 21.6 |
| | D1% | 21.3 | 21.6 | 21.9 | 22 | 21 | 21.4 | 21.5 | 22.2 |
| Spinal Cord | Dmean | 0.1 | 0.1 | 0.2 | 1.7 | 0.2 | 0.1 | 0 | 1.6 |
| | D5% | 0.2 | 0.2 | 0.3 | 5 | 0.3 | 0.3 | 0.1 | 5.5 |
| | D1% | 0.3 | 0.3 | 0.4 | 5.5 | 0.5 | 0.3 | 0.2 | 6 |
| Heart | Dmean | 0.1 | 0.1 | 0.1 | 0.6 | 1 | 1.8 | 0.7 | 2.6 |
| | D50% | 0.1 | 0.1 | 0.1 | 0.1 | 0.2 | 0.2 | 0.1 | 2.3 |
| | D20% | 0.1 | 0.1 | 0.1 | 1.7 | 0.3 | 0.4 | 0.1 | 4.8 |
| | D5% | 0.2 | 0.2 | 0.2 | 2.6 | 9.1 | 19 | 2.9 | 7 |
| | D1% | 0.7 | 0.3 | 0.3 | 3.1 | 10.2 | 20.4 | 19.1 | 9.4 |
| Right Lung | Dmean | 6.4 | 7.9 | 5 | 5.6 | 3 | 0.1 | 5.4 | 2.2 |
| | D50% | 4.8 | 0.6 | 0.2 | 3.5 | 0.2 | 0.1 | 0.1 | 2.6 |
| | D20% | 10.8 | 20.4 | 16.8 | 9.9 | 9.1 | 0.1 | 18.3 | 3.9 |
| | D5% | 20.4 | 21.5 | 21.3 | 20.2 | 10.8 | 0.2 | 21.4 | 4.9 |
| Left Lung | Dmean | 3.3 | 0.1 | 6.7 | 1.9 | 8.6 | 9.7 | 7.3 | 8.9 |
| | D50% | 1 | 0.1 | 0.4 | 1.9 | 9.3 | 6.7 | 0.2 | 8.2 |
| | D20% | 6.8 | 0.1 | 20.4 | 3.4 | 18.4 | 20.4 | 20.2 | 18.2 |
| | D5% | 10.6 | 0.2 | 21.9 | 4.7 | 20.8 | 21.3 | 21.4 | 21.4 |

treatment. Although consideration should be made when deciding on whether a right or left lung model is appropriate for a study's application.

One outcome of note was that, despite previous reports of 100% tumor take rates [11, 12, 18], we were only able to validate a rate of 80% tumor take rate with our protocol irrespective of right or left lung implantations. Our reported take rate could differ from other reported studies due to specifically the H1299 cell line having a different growth characteristic, as studies using other NSCLC cell lines have reported different rates of localized tumor take [12, 18]. Our different take rate could also be due to the more stringent criteria we use to deem a tumor model as having an RT treatable tumor. One such criterion is that tumor growth must be identified to be in the lung. Previous studies do not distinguish between tumor growth inside the lung and tumor growth external to the lung but inside the thoracic cavity [17]. However, Jarry and colleagues did observe that NSCLC orthotopic implantation in the lung, either through intrathoracic or intratracheal means, often resulted in the development of tumors around the lung with reduced deep infiltration of tumor cells deep in the lung [32]. In fact, BLI imaging studies ignore this distinction as the imaging modality cannot differentiate between a lesion which is inside or external to the lung. For our protocol, conformal RT treatable tumors must be developed inside the lung. This is because tumor growth inside the thoracic cavity and outside the lung contributes to the overall tumor burden of the animal and would not be actively treated by the conformal external beam RT. Thus, these thoracic tumors can affect outcomes

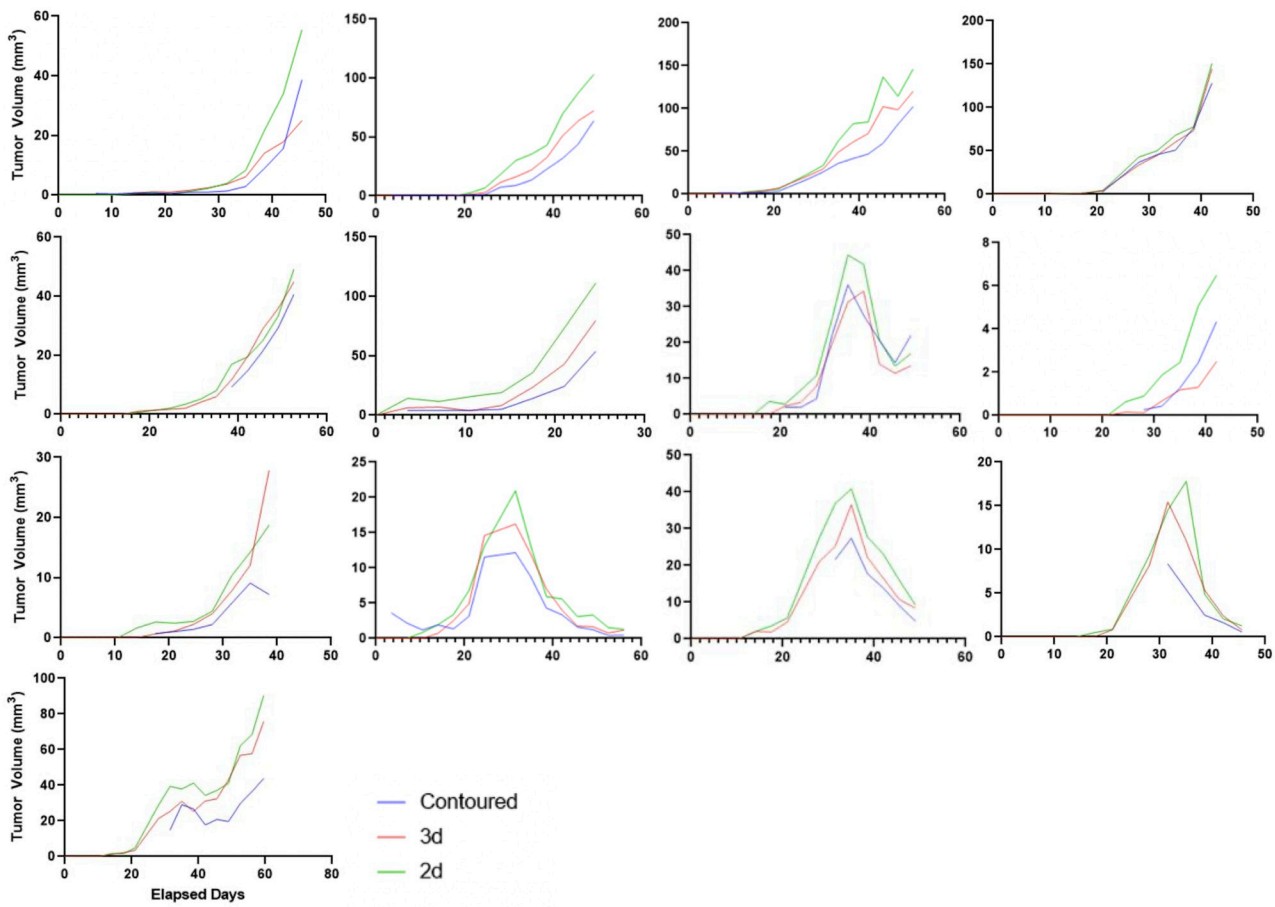

**Fig 10. Plots of tumor volume tracking for 13 contoured sample tumors featuring contoured (blue line), 3D approximated tumor volume (red line) and 2D approximated tumor volumes (green line).** The x-axis represents elapsed days while the y-axis expresses tumor volume in mm$^3$. The plots illustrate consistent tumor volume overestimation for 2D approximated tumor measures while that of 3D more closely approximated the contoured volume.

of animal survival and overall animal condition as a confounding factor. We theorize that this is a primary reason why, despite significant target tumor control, animal survival between RT and CTRL animals remains similar.

Beyond lung external or internal tumor location as a criterion, we also attempted left and right lung implantations. Tumors developed in the right or left lungs did not result in any noticeable differences in tumor take rates. However, there is a noted increased presence of external lung thoracic lesions at animal endpoints, whether due to reflux of implanted cells from the lungs during surgical injection or from metastasis. Prospective studies should be aware of the three-lobule structure of the right lung in comparison to the single lobule structure of the left. The multilobe structure, containing multiple lobe-dividing fissures and smaller lobe targets, means that there is a greater potential for injections to deposit tumor cells into the thoracic cavity rather than the lung, forming tumors external to the lung. As a result, the multi-lobe structure increases the precision required for successful implantation into the lung. These challenges led to missed injections during our pilot phase with right lung implantations but were avoided during the left lung implantations due to the left lung's single lobe structure. However, successful implantations into the right lung have the benefit that they can yield

tumors which are restricted to just the middle or lower lobe allowing for easier and cleaner separation of tumor engrafted lung tissue from normal tissue for downstream immunohistochemistry or genetic analysis applications. In general, animal survival, tumor growth and animal condition throughout the study did not vary between left or right tumor models.

With our study we've also reported on tumor growth characteristics for the H1299 cell line in our orthotopic model. In our observations, the growth period leading up to treatable tumors that are within our set volume requirements, is variable despite efforts to control it. Using different quantities of tumor cells during injection in our pilot experiments did not correlate with changes in tumor growth rates (S5 Fig). We did find that a formulation featuring 400,000 cells in 20 μL leads to a consistent take rate between trials. The limitation of our protocol to 30 days after RT treatment, is due to tumor response being observable within 2 weeks post RT treatment (Fig 5) and animal survival being within the 30-day range (Fig 7). Mice showing complete remission from treatment were able to live past 30 days without any observed complications. We had also attempted the same surgical protocols with the more commonly used A549 NSCLC cell line, injecting 60k cells, during pilot experiments. However, the A549 cell line did not yield singular tumor nodules and quickly progressed to diffuse disease resulting in much quicker animal mortality than with H1299 (S1 Fig). This finding supports the results of a comparative study evaluating the tumor progression and animal survival of different NSCLC cell line derived orthotopic models [18].

Animal weight is often not referred to in other studies and protocols despite our observation of the importance of weight as an indicator of animal condition. Despite C57BL/6J mice being reported to reach breeding age around 35 days and adulthood around 3 to 6 months [33], we found that during the first 30–40 days post-implantation surgery mice would reach a steady weight. This state of steady weight meant that observable rapid weight loss because of disease progression became obvious. During our study, weight loss preceded other behavioural indicators of animal distress and as such was a great indicator of animal wellbeing during follow-ups. However, for most control animals, around 30 days post-treatment, all animals will have exhibited one of the following signs of distress: 1) rapid weight loss, 2) behavioral distress, or 3) dyspnea.

In terms of prescribed RT doses, we were able to validate that 20 Gy produces significant tumor responses in both the 4-field and lateral opposed field configuration in both the right and left lung, respectively. Despite the larger involvement of the ipsilateral lung in both plans and the contralateral lung in the lateral opposed plan as well, all animals were treated without any observed complications. The bi-weekly full body CT imaging, which delivers approximately 5 cGy per imaging session given our image resolution and imaging energy [34], does not deposit enough dose to result in significant tumor control despite the cumulative dose being of a significant amount [35]. Despite previous studies suggesting that repeated CT imaging should be avoided [11], in our study, we can validate that repeated exposure given our imaging parameters will not lead to significant tumor control. Normal tissue complications and acute adverse events were also not observed in our animal model. Our observation accords with similar observations in a study featuring a high dose rate and high peak dose microbeam RT protocol [36] and an alternative study demonstrating fractionated 2 Gy x 25 partial-heart RT did not lead to acute complications within the first 12 months post-RT [37].

Tumor quantification is an especially difficult challenge in the case of CT imaging and orthotopic lung cancer models. With BLI imaging, tumor size has been correlated with signal intensity in many pre-clinical studies [11]. However, to capture exact tumor volumes in an orthotopic model, there is no choice but to use clinically applied, non-invasive methods of tumor measurement. Contouring gross tumor volumes in efforts to quantify exact volumes remains difficult because it is time consuming and made more so given the large quantities of

animals required in pre-clinical studies. As such, approximating tumor volumes through minimal measurements is key to an effective means of quantification. In our assessment of quantification methods, we found that the 3D method of measurement provides the best tumor volume approximation. While all three methods of quantification provide a robust quantitative metric to evaluate tumor changes overtime, and can particularly capture fluctuations in volume well, only 3D approximation provides the closest absolute volume measurements to contouring. Furthermore, quantification methods requiring the identification of a longest diameter is difficult as most CT imaging only generate a transversal, sagittal and coronal cross-section. Reproduction in these three planes may not be adequate to evaluate the true longest diameter. As such the 3D ellipsoidal approximation can bypass the need for not only contouring but also 3D reproductions of the tumor volume to determine an accurate longest diameter measure.

Despite the demonstrated effectiveness of our model, there are also limitations to be cognizant of. Currently, implantations which have deposited mainly in the thoracic cavity but external to the lung cannot be detected. The method of verifying injection deposition using CT and Omnipaque contrast can only confirm the success of injection into lung parenchyma but cannot confirm that the injection has been deposited into the thoracic cavity nor can it confirm dissemination. Furthermore, identification of exceedingly small lesions can only be done retroactively, whereby a series of images confirming tumor growth is followed to the earliest appearance in imaging. The current methodology assumes that if there is no identified contrast media in the lung parenchyma after injection, then the injection was deposited into the thoracic cavity. And while tumor growth in the thoracic cavity may not be an issue for studies featuring BLI quantification or those that feature total lung irradiation or are evaluating systemic therapies, these lesions located in the thoracic cavity are typically not visible in CT imaging until they reach exceedingly large sizes, until metastatic disease begins to appear within the lung parenchyma, or until the animal is exhibits severe distress behaviours. This means treatment of these lesions using a conformal external beam and planning around an identifiable tumor volume becomes impossible. Within our study, the 20% failed injections were comprised of these animal models. Thus, future studies requiring clearly delineated parenchymal lesions should be aware of the added difficulty of achieving this outcome. Another limitation of our work is that it did not focus on producing any data on acute or chronic organ tolerances in our specific pre-clinical model. While we did not observe toxicities in the acute phase, we cannot be certain that there were no acute toxicities which could be mild but relevant to studies in other organ systems. Furthermore, being that the follow-up time-period is only 30 days, we also do not have data on long term outcomes and appearances of adverse events, such as the development of fibrosis. Future work is needed to establish the relationship between dose deposition and degree of tumor control in mouse models. Currently, there is no consensus on what an appropriate treatment dose for pre-clinical tumors is, due to dosimetry being not well reported in conventional literature, and as a result it is difficult to translate pre-clinical findings. Within the outcomes of our study, we were able to provide evidence that 20 Gy single fraction was a well-tolerated dose in our nude mice model featuring H1299 orthotopic tumors.

In terms of treatment planning, we translated and applied clinical treatment techniques and workflows to this mouse model. Mice were initially CT imaged, treatment planning conducted on those same CT images, imaging is captured through CBCT at different points during treatment and then subsequent follow-up are conducted also through CT imaging. The adaption of the clinical workflow is time efficient and can be well applied to the treatment of animal models despite the size differences. A 5 mm diameter field was chosen due to its ability to offer conformal treatment plans while allowing the necessary coverage for our tumor volume criteria. Larger fields, of 1 cm diameter, often covered the entirety of the lung while

smaller fields, such as 2.5 mm, could only provide coverage for tumors which cannot be confirmed in the current imaging parameters due to the limits in imaging resolution.

## Conclusion

We proposed and reported on and validated the efficacy of an orthotopic mouse model of NSCLC featuring H1299 NSCLC cell line, in both the left and right lung, using a non-invasive computed tomography (CT) imaging of tumor quantification. We were able to demonstrate that both left and right lung models performed similarly, but with important differences that are relevant to studies looking to apply an orthotopic model to research requiring clearly delineable and conformal RT treatable lesions. We also validated different treatment planning for the delivery of a single ablative 20 Gy dose in both right and left lung tumor models. And ultimately, we validated an effective 3-dimensional tumor approximation method which allows similar tumor quantification performance to tumor contouring while saving more time.

## Supporting information

**S1 Fig. CT imaging of right lung tumor growth progression for representative H1299 (top row) and A549 (bottom row) implanted animals.** Each column represents a different time point, from the left the time points were post-op (far left column), imaging at first lesion which was two weeks after surgery (2nd column from the left), 3) two weeks after first lesion (2nd column from the right), and 4) one week after previous time point (far most right column). The presence of the H1299 lesion is indicated by an orange arrow.
(TIF)

**S2 Fig. Sequential CT imaging in 3.5-day intervals exhibiting a shrinking density which resolved after 14 days (5th image from the left).**
(TIF)

**S3 Fig. Plot of the growth of surgical site density post-surgery over the first 3 weeks.** A stagnation of lesion growth occurs at the end of the first week in our population of 67 recorded surgeries. Post 2 weeks, the lesions are identified as tumors as they can be traced back from instances of confirmed tumor imaging retroactively. Error bars are indicative of standard error.
(TIF)

**S4 Fig. Contour of the heart (orange dashed line) with the unclear delineated boundary between the heart and the thymus (shorted segment between the two orange arrows).** The delineated boundary between the heart and thymus was based an approximation of the heart shape as densities between the heart and thymus cannot be visually differentiated.
(TIF)

**S5 Fig. Tumor growth rates of different numbers of H1299 cells injected.** The tumors depicted in this plot were all right lung tumor models and were developed within 5 different animals. There was no correlation between increased number of injected tumor cells and increased tumor growth.
(TIF)

## Author Contributions

**Conceptualization:** Li Ming Wang.

**Data curation:** Li Ming Wang, Ranjan Yadav, Monica Serban.

**Formal analysis:** Li Ming Wang, Ranjan Yadav.

**Funding acquisition:** Norma Ybarra.

**Investigation:** Li Ming Wang.

**Methodology:** Li Ming Wang, Ranjan Yadav, Monica Serban, Osvaldo Arias, Jan Seuntjens.

**Project administration:** Li Ming Wang, Jan Seuntjens.

**Resources:** Jan Seuntjens, Norma Ybarra.

**Software:** Li Ming Wang.

**Supervision:** Monica Serban, Jan Seuntjens, Norma Ybarra.

**Validation:** Li Ming Wang.

**Visualization:** Li Ming Wang.

**Writing – original draft:** Li Ming Wang.

**Writing – review & editing:** Li Ming Wang, Ranjan Yadav, Monica Serban, Osvaldo Arias, Jan Seuntjens, Norma Ybarra.

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
