## [Decision Letter · Decision Letter 0]

27 Jan 2023

PONE-D-22-27036Validation of an orthotopic non-small cell lung cancer mouse model, with left or right tumor growths, to use in conformal radiotherapy studiesPLOS ONE

Dear Dr. Wang,

Thank you for submitting your manuscript to PLOS ONE. After careful consideration, we feel that it has merit but does not fully meet PLOS ONE’s publication criteria as it currently stands. Therefore, we invite you to submit a revised version of the manuscript that addresses the points raised during the review process.

Please address the minior comments of both reviewers point by point, and espcially improve the figure quality as brought up by the first reviewer and add necessary details of experiments as requested by the second reviewer.

We look forward to receiving your revised manuscript.

Kind regards,

Zhentian Wang, Ph.D.

Academic Editor

PLOS ONE

Journal Requirements:

Reviewers' comments:

Reviewer's Responses to Questions

**Comments to the Author**

1. Is the manuscript technically sound, and do the data support the conclusions?

Reviewer #1: Partly

Reviewer #2: Yes

2. Has the statistical analysis been performed appropriately and rigorously? 

Reviewer #1: Yes

Reviewer #2: Yes

3. Have the authors made all data underlying the findings in their manuscript fully available?

Reviewer #1: Yes

Reviewer #2: Yes

4. Is the manuscript presented in an intelligible fashion and written in standard English?

Reviewer #1: Yes

Reviewer #2: Yes

5. Review Comments to the Author

Reviewer #1: Authors describe use of orthotopic lung cancer mouse model as a tool to examine effects of radiotherapy (RT). Orthotopic mouse models have clear advantage as models that place tumors in its native environment. Authors argue that computed tomography (CT) imaging could be utilized to accurately measure tumor size following RT in orthotopic mouse models of lung cancer. Using H1299 NSCLC cell line, authors implant cells in both right and left side of the lung of the recipient mouse, perform CT based imaging, and RT.

Overall this study contributes important information about mouse models that are used in pre-clinical work in NSCLC research field. Improving some of the Figures and providing clarifications (please see below) would render this manuscript fit for publication.

Some points and questions:

1) Figures 1 and 2 look little bit fuzzy, it would be helpful to have a slightly more clear pictures. Also, if possible it would be great to have a schematic drawing of major anatomical landmarks seen in Figure 1 and 2.

2) Figure 4: green arrow is very small; please consider increasing size of the arrow. I also don’t see red lines that are described in the Figure Legend.

3) Does including Omnipaque in the injection mix with the cells and Matrigel influence resolution of CT scans?

4) Figure 7: on the graph on the right – does the “days elapsed” refer to post RT or post-surgery?

5) Figure 8: right side of the figure shows that survival was similar between control and RT treatment groups. However, Figure 5 shows that tumor size ratio was much higher in control group. Could you please clarify if Figure 5 and Figure 8 show the same treatment regiment?

6) Figure 10: please label axes on at least one of the graphs.

Reviewer #2: The authors describe a preclinical orthotopic tumor model of non small cell lung cancer in immunocompromised mice, which they then use for a conformal radiotherapy treatment study. The model is very elaborate and relevant. The manuscript is well written and the experiments and results are well described.

- In the introduction, the authors are very critical of the use of luminescence for imaging, but others have shown that it still allows monitoring the response of an orthotopic tumor to therapy and the spread of cells in the body due to a very high detection sensitivity (Jarry et al., 2021).

It is true as mentioned by the authors that transfection of cells with a plasmid increases the immune response to the generated tumor (plasmids limiting this effect are however available); but, to date, there is no syngeneic model of lung adenocarcinoma so the question does not arise. In addition, the authors use H1299 cells (human) injected into immunodeficient mice to prevent rejection. The immune system cannot be studied in this context.

- As mentioned by the authors CT is a good imaging method to evaluate the volume of a tumor. However, the authors should also mention that it is a method not allowing to detect a small number of cells and limited dissemination.

- The authors do not explain why they used a single dose of 20Gy to treat the tumors when this is not the protocol used in the human clinic?

- The authors do not clearly mention how long after injection the tumors are visible by CT in mice? What is the minimal volume that is detected by CT?

- The authors mention that tumor uptake is about 80%; however, they do not say how this was assessed, is it only by CT imaging or was this also verified by resection and tissue analysis for mice with negative imaging?

- The following sentence is a bit odd: "approximately 4 in 10 animals that had right injections were found to have external lung lesions". The authors must know precisely how many animals have tumor development outside the lung (which is a common finding with this implantation technique). In addition, other authors have observed and mentioned this phenomenon (Jarry et al., 2021) contrary to the authors' claims. This should be mentioned in the manuscript.

- The authors do not say whether when the mice were euthanized the lungs were removed and the tissues analyzed to correlate with the imaging findings, which would have been an excellent control.

- In Figure 5, the authors show a very large effect of radiotherapy treatment on tumor volume (reduction) observed by imaging; while at the same time the tumor volume in untreated mice increases very strongly. However, Figure 8, this does not seem to have an effect on the survival of the mice with a very similar effect with or without treatment. How do the authors explain this result? This should be explained in the manuscript.

6. PLOS authors have the option to publish the peer review history of their article (what does this mean?). If published, this will include your full peer review and any attached files.

Reviewer #1: No

Reviewer #2: No

---

## [Author Response · Author response to Decision Letter 0]

17 Mar 2023

Dear Dr. Wang and Reviewers,

Thank you very much for all the time and effort in reviewing the manuscript and providing to us your comments and suggestions. They are very much appreciated. Below, in point form, we’ve added rebuttals to points brought about by reviewers:

Reviewer 1:

1. We’ve replaced the images with higher resolution ones. We attempted a schematic drawing however, we found that a schematic, when not overlayed onto an image, to be equally as difficult to decipher. We ultimately added the original anatomical landmark labelling to the new updated images. We find that this is the clearest representation and labelling of anatomical landmarks.

2. We’ve enlarged the arrows and red lines so that they are more visible against the background CT imaging. 

3. In this case it does not. Omnipaque functions as an iodine based radiocontrast agent which enhances visibility of structures which uptake the contrast agent. Omnipaque is also a radiocontrast agent commonly used in the clinic and as such is designed to not adversely affect image quality. Matrigel is an extracellular matrix (ECM) preparation which solidifies at body temperature and provides an appropriate surrounding for the cancer cells to grow. Neither of these agents affect CT resolution which is based on the size and number of detector elements, the size of the X-ray focal spot, and the source-object-detector distances. The latter two of these three characteristics have been optimized as a part of regular equipment maintenance of the small animal irradiator system outlined in our protocol, while the first of these components does not change. Furthermore, we recognize that it could be plausible that the agents themselves contribute to imaging artifacts which affect the quality of the captured images. However, neither agent used in our protocol contain metals with high atomic numbers and, as such, will not cause possible artifacts which affect image quality. 

4. Days elapsed refers to number of days since initial tumor implantation. We have made additions to the figure caption and to the figure axis label for Figure 7 to specify this information. 

5. The two images do indeed show the same treatment regimen. A modification to Figure 8’s caption was made to highlight this. 

6. Axis labels were added to the last graph in the figure. Graphs were also rearranged to ensure better alignment.

Review #2

1. Our intentions with the passage in the introduction were to highlight the limitations of bioluminescent imaging specifically when applied to studies involving external beam radiotherapy, such as the study of radiosensitizer responses or combined therapies on solid single nodule tumors common to NSCLC. In the context of these studies, treatment planning is dependent on clear 3D imaging reconstructions of tumors. We added an additional sentence to clarify this point. We agree with the reviewer that bioluminescence imaging does indeed have high detection sensitivity, but this property is disadvantageous in studies requiring tumor treatment planning as it is unable to clearly delineate exact tumor volumes. In our introduction, we were trying to highlight that CT imaging is the better method for applications involving external beam radiotherapy treatment planning in an orthotopic animal model. However, we felt that it was also important to highlighting the limitations of bioluminescence, which is a very popular and useful imaging method that has seen application in orthotopic models. 

With regards to immune response, while we agree with the reviewer that studies featuring the adaptive immune responses, which features T-cell activity is not possible, as the athymic nude mice utilized in our model lacks T-cells, we do feel that these models can be used in studying innate immune responses. And while studies featuring the innate immune response and their response to transfected cells are limited, we wanted to still mention the possibility of interaction as we believe it is still valuable to be aware of possible innate immune responses even in adaptive immune compromised mice models. To that end we’ve added a sentence specifically clarifying that the effect most relevant for immunocompromised mice is effects to the innate immune system. 

2. We have added additional clarifying statements regarding the limitation of CT imaging with regards to tumor size and dissemination. We agree the lack of verifying dissemination of cancer cells is a limitation to our imaging technique. Our protocol featuring CT imaging can only confirm the presence of successful injection into the lung parenchyma, it cannot confirm dissemination to other parts of the body. Furthermore, we noted that the identification of exceedingly small lesions can only be identified retroactively, requiring a series of follow-up imaging which provides a growth trajectory that can be followed back in time to identify the smallest possible lesion. While this is not the most effective way, this method of tumor identification was a recommendation resulting from our collaboration and consultation with a qualified clinical radiologist and replicates how clinicians approach diagnoses using CT imaging for lung lesions. 

3. Human clinical protocols utilize much higher cumulative doses which aren’t relevant in mouse models and cannot be delivered by the small animal irradiation system. This is due to differences in dose rates and the treatment planning for small rodents being not as efficient or precise compared to clinical planning and delivery. As such it is not possible to extrapolate pre-clinical doses from clinical protocols and vice-versa. As an example, in clinical protocols for stereotactic body radiotherapy (SBRT) for the treatment of stage II non-small cell lung cancers (NSCLC), the prescribed dose is typically 20 Gy per fraction in 3 fractions resulting in a cumulative dose of 60 Gy. However, in clinical applications of SBRT the treatment of stage II NSCLC tumors are, proportionally to the lung, are very small. As such SBRT field sizes are also very small allowing for healthy lung tissue sparing. However, due to the size of the mice lung, irradiation fields often cover a majority of a single lung. As such there is not much tissue sparing. As such, we did not attempt to base our preclinical treatment protocol on that which is currently used in the clinic.

However, despite clinical protocols being difficult to directly adapt in a pre-clinical setting, there are pre-clinical studies that have been conducted which demonstrate that our applied 20 Gy dose is well tolerated in rodents. We’ve added the citations as well as a clarifying statement in the methods to explain that our choice of 20 Gy was based on previous studies which showed that there is good rodent tolerance for the dose. 

4. Tumors remain visible if they are 1) located in the lung, 2) maintain a shape distinctly different from surrounding vessels and soft tissue structures and 3) can be interpreted over time (meaning there are more than just one static image at one time point and that changes in tumor can be appreciated and followed). Throughout our 30-day follow-up period the tumor remained visible unless macroscopic evidence is not observed. This is due to the tumor being a denser soft tissue density among the lower density lung parenchyma. As mentioned in the results, there was an overlap period between the disappearance of the initial high density, likely due to acute inflammation or residual contrast media, and the appearance of the lesion. This overlap makes it difficult to determine the smallest lesion. Despite this overlap we were able to detect lesions that were as small as 0.1 mm3 as can be appreciated on our plot in Figure 5. 

5. This was a combination of CT and resection. An additional methods section was added to detail the process of visual validation at the time of animal euthanasia. 

6. We’ve changed the phrasing to the external lung tumor ratio that were presented in our study with the exact ratios that we observed. We’ve also added mentioned of the findings of Jarry et al., specifically with regards to their observation of tumor growth outside the lung and observations of reduced deep penetration of implanted cancer cells. 

7. As the reviewer had also alluded to in point 5, we have since added the much-needed section regarding animal euthanasia and identification of region of lesion growth. Tissues were not analyzed by histology, but visual identification was conducted to validate the results from the imaging section.

8. Due to the instances of lung external tumors and the application of conformal radiation treatments featuring external beams, it is difficult for us to ascertain the true effect of tumors on animal that are not visible by CT, such as the lung external tumors. Furthermore, our model is also limited in its capacity to detect metastasis. The combination of non-visible tumors and possible metastatic sites can affect animal survivor despite target tumor control. Further study on out of field tumors is warranted for an orthotopic model like ours. We have added a statement which relates our mention of lung external tumors to the outcomes seen in Figure 5 and Figure 8.

Editorial Points:

1. We have made a change to our financial disclosure. While we did include in the acknowledgements, we also added it to the cover letter. We understand that the financial disclosure is likely not appropriate for the acknowledgements, as was specified by the formatting documents, but we kept it there in case, as per tradition. We will remove it if necessary. 

2. Formatting made to the manuscript to adhere to PLOS One’s formatting requirements

3. Reintegration of the Supplemental information into the manuscript document.

4. Uploads of the Supplemental figures separately.

---

## [Decision Letter · Decision Letter 1]

29 Mar 2023

Validation of an orthotopic non-small cell lung cancer mouse model, with left or right tumor growths, to use in conformal radiotherapy studies

PONE-D-22-27036R1

Dear Dr. Wang,

We’re pleased to inform you that your manuscript has been judged scientifically suitable for publication and will be formally accepted for publication once it meets all outstanding technical requirements.

Kind regards,

Zhentian Wang, Ph.D.

Academic Editor

PLOS ONE

Additional Editor Comments (optional):

Reviewers' comments:

Reviewer's Responses to Questions

**Comments to the Author**

1. If the authors have adequately addressed your comments raised in a previous round of review and you feel that this manuscript is now acceptable for publication, you may indicate that here to bypass the “Comments to the Author” section, enter your conflict of interest statement in the “Confidential to Editor” section, and submit your "Accept" recommendation.

Reviewer #1: All comments have been addressed

Reviewer #2: All comments have been addressed

2. Is the manuscript technically sound, and do the data support the conclusions?

Reviewer #1: Yes

Reviewer #2: Yes

3. Has the statistical analysis been performed appropriately and rigorously? 

Reviewer #1: Yes

Reviewer #2: Yes

4. Have the authors made all data underlying the findings in their manuscript fully available?

Reviewer #1: Yes

Reviewer #2: Yes

5. Is the manuscript presented in an intelligible fashion and written in standard English?

Reviewer #1: Yes

Reviewer #2: Yes

6. Review Comments to the Author

Reviewer #1: (No Response)

Reviewer #2: The authors responded satisfactorily to all comments. The presented study has some weaknesses, but the work is sound and the authors present their work honestly. Finally, this work is of great interest to the scientific community.

7. PLOS authors have the option to publish the peer review history of their article (what does this mean?). If published, this will include your full peer review and any attached files.

Reviewer #1: No

Reviewer #2: No

---

## [Editor Report · Acceptance letter]

3 Apr 2023

PONE-D-22-27036R1 

Validation of an orthotopic non-small cell lung cancer mouse model, with left or right tumor growths, to use in conformal radiotherapy studies 

Dear Dr. Wang:

I'm pleased to inform you that your manuscript has been deemed suitable for publication in PLOS ONE. Congratulations! Your manuscript is now with our production department. 

Kind regards, 

on behalf of

Prof. Zhentian Wang 

Academic Editor

PLOS ONE